# Control of Neonatal Diarrhea in Piglets with Reduced Antibiotic Use by Application of a Complementary Feed—A Randomized Controlled Farm Trial

**DOI:** 10.3390/vetsci12010042

**Published:** 2025-01-10

**Authors:** Klaus K. Sall, Leslie Foldager, Charlotte Delf, Sigurd J. Christensen, Michael N. Agerley, Kristian T. Havn, Carsten Pedersen

**Affiliations:** 1Sall&Sall Advisors, DK-8220 Brabrand, Denmark; 2Newtrifeed ApS, DK-6372 Bylderup-Bov, Denmark; sigurd@newtrifeed.dk (S.J.C.); michael@porcus.dk (M.N.A.); kristian.havn@porcus.dk (K.T.H.); 3Department of Animal and Veterinary Sciences, Aarhus University, DK-8830 Tjele, Denmark; leslie@anivet.au.dk; 4Bioinformatics Research Centre, Aarhus University, DK-8000 Aarhus, Denmark; 56360 Tinglev, Denmark; 6Porcus Pig Veterinarians, DK-5220 Odense SØ, Denmark; 7Pedersen Nutrition, Ltd., Shaftesbury GB-SP7 9QG, UK; carsten@pedersennutrition.co.uk

**Keywords:** *Clostridium perfringens*, tannin extract, neonatal diarrhea, O-Nella-Protect, polyphenole, randomized controlled trial, RCT, *Rotavirus*, RWA, piglet, *Castanea sativa*

## Abstract

New programs and tools are needed in pig production as societal pressure to reduce antibiotic use leads to increasingly restrictive regulations. Neonatal diarrhea among newborn piglets is one of the key challenges in reducing antibiotic consumption. In this 4-week farm trial, we compared the use of a complementary feed containing a tannin extract derived from the wood of sweet chestnut with the standard antibiotic treatment used on the farm. All 18 litters in the study required treatment due to a high incidence of diarrhea among the piglets. The piglets in the nine litters of the control group received an oral antibiotic treatment. In contrast, the nine litters in the test group were initially treated with the complementary feed containing hydrolyzable tannins. If diarrhea persisted after three days, piglets were treated with antibiotics. The results showed that antibiotic consumption for diarrhea in the test group was reduced by 84% compared to the control group. Mortality rates were identical in both groups, with six piglets dying in each group. We conclude that the complementary feed containing a tannin extract has the potential to significantly reduce antibiotic use for neonatal diarrhea in newborn piglets while maintaining similar health outcomes.

## 1. Introduction

While farm managers want to avoid diarrhea among the piglets, society wants farmers to avoid the use of antibiotics. New regulations have within the past two decades put pig production under increasing pressure to reduce the use of antibiotics due to increasing numbers of antibiotic resistant bacteria and the danger that such bacteria should be transferred to humans [1,2,3]. In response to this development, new production concepts have developed like “Raised Without Antibiotics” (RWA) [4,5]. To reduce the consumption of antibiotics or to realize RWA concepts successfully, there is a need to adapt management, feed composition, immunization programs, and develop new feed products that will support the natural immune system and the microbiota of pigs [6].

During the past 30 years, several pathogens have been associated with neonatal diarrhea in piglets, including *Escherichia coli*, *Clostridioides difficile*, *Clostridium perfringens*, *Rotavirus type A*, and *Rotavirus type C* [7,8,9,10,11,12,13,14,15]. On the other hand, a Danish study that compared diarrhetic with non-diarrhetic piglets on four farms concluded that no specific pathogen could be identified as the cause of neonatal diarrhea [16].

It is a common and documented observation that piglets born from gilts have a higher incidence of neonatal diarrhea [17,18,19]. This has been linked to a lower production of colostrum and a lower level of immunoglobulin G and/or A in the colostrum of primiparous sows than in the colostrum of multiparous sows [18,20,21].

Hydrolysable tannins are widely used as an additive in feed for pigs to modulate intestinal health and enhance growth performance [22,23,24,25]. Among hydrolysable tannins, ellagitannins stand out, as research indicates that they have antioxidant, anti-inflammatory, antibacterial, antiviral, and astringent capacity, therefore having a stabilizing effect on pig physiology and intestines [26,27].

The aim of this farm test was consequently to investigate if an oral application of a complementary feed product that contained the hydrolyazable tannins from sweet chestnut to newborn piglets would treat neonatal diarrhea and consequently curb the use of antibiotics that are often used to treat this condition.

## 2. Materials and Methods

The Groenbo test farm holds about 2000 sows and produces about 70,000 pigs a year. The gilts in the trial were Landrace × Yorkshire (L × Y), and the male line was Duroc (D). Gilts were bought from another farm and inseminated at the Groenbo farm. The farm had been remediated for Porcine Reproductive and Respiratory Syndrome (PRRS) virus, but it had been reinfected and therefore had a status as Specific Pathogen Free (SPF) + PRRS [28], and it was in a re-remediation process. Gilts had been vaccinated against PRRS1 and PRRS2, as well as against *Rotavirus type A*, but not against any bacterial pathogen.

The standard treatment of neonatal diarrhea at the farm was the antibiotic Parofor^®^, which contains 200 mg/mL of aminoglycoside paromomycin sulfate (corresponding to 140 mg/mL paromomycin) and was sold by Huvepharma NV, 2600 Antwerpen, Belgium. This antibiotic was therefore used in the initial treatment of diarrhea in the control group. It was dosed orally at 0.15 mL/kg BW piglet/day for two consecutive days [29]. The same aminoglycoside was used for follow-up treatments of diarrhea when it occurred after the initial treatment in both test and control groups.

Two other antibiotics used for piglets were registered during the trial. Streptocillin^®^ is the standard treatment against joint infections, as well as other local or systemic infections. This product is a combined antimicrobial, which contains benzylpenicillin (200 mg/mL) and dihydrostreptomycin (250 mg/mL). It was administered as intramuscular injection (IM) at 0.1 mL/kg BW/day. The product is marketed by Boehringer Ingelheim Animal Health Denmark A/S, 2300 Copenhagen S, Denmark [30]. The antibiotic Flordofen^®^ is the standard treatment against respiratory infections. This antibiotic contains florfenicol (300 mg/mL); it was dosed IM twice at 0.05 mL/kg BW with 48 h in between injections. The product is marketed by Dopharma Research B.V., 4941 VX Raamsdonksveer, The Netherlands [31].

The product O-Nella-Protect was used to treat neonatal diarrhea in the test group and is registered as a complimentary feed. The main feed ingredients are glycerol (75%); vegetable charcoal; mono-, di-, and triglycerides of butyric acid; and a feed additive extract from sweet chestnut wood (*Castanea sativa* Mill.) containing 70% hydrolysable tannins high in ellagitannins. All ingredients are EU-approved feed ingredients and additives [32,33]. The product was dosed with one ml per piglet per day for three days. The product is marketed by Newtrifeed ApS, 6372 Bylderup-Bov, Denmark.

The consumption of antibiotics and death rates were measured in 20 litters totaling 284 piglets born by gilts distributed to 10 test litters—totaling 143 piglets—and 10 control litters—totaling 141 piglets. The piglets were Landrace × Yorkshire × Duroc (L × Y × D).

The gilts were distributed at random in the enclosed section before farrowing so that the resulting 10 litters to the left of the central access path constituted the control group and the 10 litters to the right constituted the test group. The number of piglets in litters were adjusted to fit the capacity of the sow within the first 24 h after parturition.

Neonatal diarrhea is here defined as symptoms of diarrhea during the first 1–5 days after birth at a time when the sole feed for all piglets is colostrum, followed by milk from the dam. Cases of diarrhea were defined equally for the two groups as slightly sunken pigs that were dirty by the tail. Cases were included when treatments were performed. Treatments of diarrhea were metaphylactic, as the full litter was treated when the first diarrhetic piglet in the litter was observed.

All management, procedures, and registrations used for this 4-week study followed the standard procedure of the farm, with no additions except for the alternative product for treating diarrhea in the test litter—as agreed in the written protocol of the farm trial.

Test litters: In case of symptoms of neonatal diarrhea, the full litter was administered O-Nella-Protect in a primary treatment period for three consecutive days (p1). In case of continued symptoms of diarrhea on the fourth day after initiation of O-Nella-Protect administration or later recurrence of diarrhea in the follow up period, the full litter of piglets were treated for two consecutive days with paromomycin according to the prescription of the veterinarian (p2).

Control litters: In case of symptoms of diarrhea, the full litter of piglets was treated with one dose of paromomycin for two consecutive days in a primary treatment period according to the prescription of the veterinarian (p1). In case of continued symptoms of diarrhea in the follow up period on the third day (or later), the piglets were treated for two consecutive days with paromomycin according to the prescription of the veterinarian (p2). No other antibiotic was used for diarrhea.

All piglets were given a standard treatment at day 3–4 against coccidia (Toltrazuril, distributed by KRKA, d.d., 8501 Novo mesto, Slovenia). All male piglets were given a standard combination treatment of anesthetic, analgesic, and antibiotics during the castration process on day 2 or 3 after birth. As these treatments were systematic, they have not been considered in the data of the trial.

The registration of animals and antibiotics was done for each litter. All applications of medicine were registered by name of the antibiotic and doses used for each day/litter.

### 2.1. Microbiological Test

Two one-day-old and two two-day-old piglets with untreated neonatal diarrhea that had been born by gilts, but which were not part of either control or test litters, were euthanized and sent for autopsy and aerobic and anaerobic microbiological identification of pathogens at the Danish Veterinary Clinical and Research Laboratory, Kjellerup, Denmark.

Fecal samples from diarrhetic untreated piglets were sent for laboratory analysis for identification of pathogens at the laboratory Miprolab, Göttingen, Germany. This laboratory has special competence in the identification and cultivation of *Clostridia* species and types. Samples were pooled before microbiological analysis.

The veterinary laboratory in Kjellerup also analyzed for antibiotic resistance in the indicator species *E. coli* according to their laboratory procedures to establish Minimum Inhibitory Concentration (MIC) standard. The breakpoints selected by the laboratory have been detailed in Pedersen et al. (2021) [34].

### 2.2. Statistical Methods

The experimental unit was the individual litter, and the response can be seen as either (1) treatment (yes/no) of the litter, (2) number of treatments applied in the litter, or (3) the number of doses administered to piglets of the litter.

For paromomycin, the treatments and number of doses administered can be interpreted either in total across the study period, only as second round of (or continued) treatment after three days of treatment with O-Nelle-Protect, or two days with paromomycin treatment (period 2). No litters received more than two rounds of paromomycin treatment.

Fisher’s exact test of independence in a 3-by-2 contingency table was applied to compare number of treatments (0, 1, or 2, with a treatment consisting of 2 consecutive days) with paromomycin between the test and control litters throughout the four weeks of study. For period 2, treatment (yes/no) was examined by logistic regression. For total number of doses throughout study period or in period 2 only, test and control litters were compared by Wilcoxon–Mann–Whitney rank sum test.

Treatment (yes/no) of the litter with florfenicol and Streptocillin, respectively, was examined using logistic regression. Moreover, the rate of treatment (doses) with florfenicol was compared between the test and control groups using Poisson regression, with the logarithm of litter size as an offset.

Results from logistic and Poisson regressions are given as odds ratios (OR) and rate ratios (RR), respectively, for the test vs. control group, with standard error (SE) and 95% confidence intervals (CIs). The difference between test and control groups was tested using chi-squared likelihood ratio tests with 1 degree of freedom (χ^2^). Results from Fisher’s exact test and the Wilcoxon–Mann–Whitney rank sum test are given as the exact *p*-value.

Analyses were carried out using the statistical software R version 4.1.2 [35] with a significance level of 5%.

### 2.3. Ethical Considerations and Approval

The study was carried out in compliance with the Danish Act on the Protection of Animals [36], which incorporates several EU regulations to protect animal welfare, including the Council Directive on the protection of pigs [37]. According to EU Directive 2010/63/EU for animal experiments, Article 1, Section 5, the trial does not fall within the scope of animal experiments [38]. According to the ’Guidelines on Application for Clinical Trials of Veterinary Medicinal Products in Animals’ (page 4), the trial did not qualify as a clinical trial under the legislation, as analysis of pharmacodynamics or pharmacokinetics was not part of the trial [39]. For these reasons, there was no further ethical approval needed for the trial.

## 3. Results

The farm test started out with 20 litters from gilts distributed to 10 litters in the test group and 10 litters in the control group. The number of litters was during the trial reduced to 18 with 9 litters in each group, as two gilts were not able to care properly for their litters, and the farm assistant therefore decided to exchange their litters with stronger litters from sows outside the trial. For this reason, the registrations from these two litters were excluded from the test.

The registered treatments in the test and control groups are presented in the below Table 1. The difference in total antibiotic treatments between the test group (394 doses) and the control group (76 doses) amounted to a reduction of 318 doses corresponding to a total reduction of 81%. Antibiotic treatments for diarrhea were reduced by 84% (*p* = 0.001; rank sum test for total number of paromomycin doses in litters), while antibiotic treatments for various infections were reduced by 45% (RR = 1.8, *p* = 0.045). Also, after the initial treatment with either the O-Nella-Protect or paromomycin, the proportion of piglets treated for diarrhea with antibiotic was double as high in the control piglets (46%) as in the test piglets (23%) (OR = 2.7, *p* < 0.001).

The consumption of antibiotics in test and control groups is illustrated below in Figure 1.

The death rate was slightly lower in the test litters at 4.69% against 4.76% in control litters. This difference was non-significant. In both the control and test groups, 116 piglets were weaned, resulting in a weaning percentage of 92.1% in the control group against 90.6% in the test group (OR = 1.2, SE = 0.54, 95% CI: 0.50–2.9, *p* = 0.68). This difference covered six dead piglets in both the control and the test groups, while a total of six piglets were removed from the sows of the test litters, but only four were removed from the sows of the control group. This difference was also statistically non-significant.

In the test group, all piglets were administered O-Nella-Protect for three consecutive days in the primary period (p1) when diarrhea appeared. In two litters (30 piglets), the diarrhea reappeared during the follow up period (2), and these were treated with paromomycin for two days (60 doses). Other infections of lungs and limbs that were treated with antibiotics resulted in 16 doses, which were specified as 12 with florfenicol and 4 with streptocillin. The total number of antibiotic doses among the piglets in the test group was therefore 76 doses.

Due to the detection of neonatal diarrhea in all litters of the control group, all litters were treated with paromomycin for two consecutive days in the primary treatment period (p1). In four litters, diarrhea continued or reappeared, and these litters were again treated with paromomycin in a follow up period of two days (p2). The primary paromomycin treatment of the nine litters resulted in 251 doses, and the follow up round of diarrhea treatment resulted in 114 doses. The combined consumption of antibiotics against diarrhea therefore amounted to 365 doses. Other infections of the lungs and limbs were treated, resulting in 29 antibiotic doses, which were specified as 26 with florfenicol and 3 with Streptocillin. The total number of antibiotic treatments among the control litters was 394 doses.

Overall, the number of times a litter was treated with paromomycin (0, 1, or 2 treatments for 2 days) differed between the test and control litters (Fisher’s exact *p* = 0.002). The odds of being treated with paromomycin after the initial treatment with either paromomycin or O-Nella-Protect were higher in the control group than in the test group (OR = 2.8, SE = 2.93, CI: 0.38–26.7), though the difference was not statistically significant (χ^2^ = 1.01, *p* = 0.31). The number of doses applied in the control litters was higher than in the test litters overall (Wilcoxon–Mann–Whitney Z = 3.02, exact *p* = 0.001) and also higher in the second round (period 2) after initial treatment but was not statistically significant (Z = 0.79, *p* = 0.43). Nevertheless, ignoring the correlation within litters, considering the 2-by-2 table of paromomycin-treated piglets (57 and 30) vs. non-treated (68 and 98) in the control and test litters, respectively, the odds would be significantly higher in the control litters (OR = 2.7, CI: 1.5–4.9, Fisher’s exact *p* < 0.001).

The rate of florfenicol treatment (doses) was higher in the control litters than in the test litters (RR = 2.2, SE = 0.77, CI: 1.1–4.5, χ^2^ = 5.50, *p* = 0.019). The odds for at least one treatment with florfenicol in the litter were also higher in control litters (OR = 2.8, SE = 2.93, CI: 0.38–26.7) but not significantly (χ^2^ = 1.01, *p* = 0.31). Note that these numbers were the same as for paromomycin in the second round, but this was a coincidence stemming from the fact that the numbers on the diagonals of the corresponding 2-by-2 tables are the same. Specifically, two test litters and four control litters were treated with paromomycin after the initial treatment (seven and five were not treated), whereas five test and seven control litters were treated with florfenicol (four and two were not). The odds for at least one treatment with Streptocillin was lower in the control than in the test litters (OR = 0.44, SE = 0.582, CI: 0.02–5.6) but not significantly (χ^2^ = 0.41, *p* = 0.52).

Considering the treatments with florfenicol or Streptocillin, all three litters receiving streptomycin also received florfenicol at some point. Thus, the odds ratios for the antibiotics used for treatment of other infections of the lungs and limbs are the same as those presented for florfenicol. Pooling of these outcomes resulted in a significant difference in the same direction as for florfenicol alone (RR = 1.8, SE = 0.57, CI: 1.0–3.5, χ^2^ = 4.02, *p* = 0.045). Assuming each piglet was treated at most once with florfenicol or streptomycin and ignoring again the correlation among piglets from the same litter, the 2-by-2 table of other antibiotics (29 and 16) vs. non-treated (97 and 112) also indicates higher odds in control than test piglets (OR = 2.1, CI: 1.0–4.4, Fisher’s exact *p* = 0.033).

The microbiological tests identified high numbers of non-hemolytic *E. coli*, as well as *Clostridium* sp. The test by Miprolab by PCR and ELISA identified *Clostridium* as an α-toxin-producing *C. perfringens*, while the result after culture proved that it was both α- and ꞵ2-producing. One laboratory found *Rotavirus type A* in the samples. The results are detailed below in Table 2.

The identified non-hemolytic *E. coli* carried antibiotic resistance towards 6 out of 15 antibiotics tested (40%) and had further developed intermediate resistance to one more antibiotic. The details are specified below in Table 3.

## 4. Discussion

It is clear that the large reduction in the consumption of antibiotic units in this farm trial reflects both a widespread registration of diarrhetic symptoms in piglets and managerial decisions about how these can be handled on a daily basis. Neonatal piglets are rather vulnerable, and neonatal diarrhea in piglets is mainly associated with lowered animal welfare, reduced productivity, and a certain level of mortality [40]. It is therefore not surprising that the test farm had little tolerance to neonatal diarrhea, as it was endemic in gilt litters during this period of remediation for Porcine Reproductive and Respiratory Syndrome (PRRS). Therefore, the full litter of piglets was treated in a metaphylactic effort at first sight of scouring in a single piglet. This approach to the treatment of diarrhea in gilt litters was the same for both the control and the test groups. The 84% reduction in antibiotic use for neonatal diarrhea achieved in the trial therefore seems relevant for farms facing similar challenges.

The gilts were allocated at random to the control or test groups, as the farm had no knowledge of the expected litter size of the individual gilts and certainly no knowledge of their ability to produce colostrum, milk volume, or tend their litters. The distribution of test and control litters to either side of the access path was done at the request of the farm manager to reduce the risk of farm assistants mixing up treatments.

Due to the high prevalence of neonatal diarrhea, the farm manager was of the opinion that it would be unethical to run an untreated control group, as a high number of piglets would be sick, and several could have died. This position was supported by the farm veterinarian who had assisted in the PRRS remediation program and who had supported in handling the problems with diarrhetic piglets.

During remediation against PRRS, strict rules for the movement of pigs were implemented according to the guideline outlined in [41]. As gilts were bought from another farm, they had not been exposed to the normal sow environment of the test farm before having the first litter, and therefore, gilts may have had no or low immunity towards various farm pathogens. As an abrupt consequence of the change of rules during remediation, a high prevalence of neonatal diarrhea occurred with an estimated 80–90% diarrhetic of piglets distributed across all litters.

After testing several antibiotics, the farm had chosen as a standard to utilize the oral antibiotic that contained paromomycin for an efficient treatment of neonatal diarrhea among piglets. For this farm trial, the use of an oral antibiotic in the control group was an advantage for two reasons: The same oral procedure of administration lowered the differences in application procedures between the test and control groups, avoiding a potential bias due to such differences. Moreover, the oral antibiotic was never used for sows, whereby a risk was avoided of misregistration of the antibiotic treatment used for either piglets or sows.

Since treatments were registered as the number of doses administered to the litter, statistical analyses were applied with the litter as the experimental unit. Regression analyses were applied where possible, e.g., as Poisson regression to compare the rate of administering antibiotics between the test and control litters or as logistic regression when comparing the odds of at least one treatment in the litter. Some responses were highly skewed or had almost all observations within a treatment group at the same value, and thus, the Wilcoxon–Mann–Whitney rank sum test was applied to avoid a doubtful distributional assumption. Likewise, for the contingency table comparing number of treatments with paromomycin, Fisher’s exact test was used to handle table cells with zero counts.

While the statistical reduction in the consumption of antibiotics is quite evident, the mechanism of action of the complementary feed is more elusive and may involve multiple factors. Three relevant aspects are in short discussed below: energy supply, microbiology, and physiological effects.

### 4.1. Energy

Based on the literature, the daily milk intake by piglets during day two to five after parturition is estimated to be 300 mL/day, growing to about 650 mL/day on average [20,42]. Likewise, the dry matter in sow milk can be estimated to be about 18.5% [42,43,44,45], so the piglets daily dry matter intake has therefore been estimated to grow from 55 to 125 g/day on average, corresponding to an average energy supply in the range of 270 kJ/piglet/day during day two growing to 600 kJ/piglet/day on day five [46].

The composition of the complementary feed was designed to deliver the hydrolysable tannins in a feed matrix suspension, as the product contains 75% of glycerol plus a low concentration of vegetable charcoal and mono-, di-, and triglycerides of butyric acid. The 1 mL of complimentary feed therefore contributed an estimated 1.15 g of dry matter/day and 20 kJ/day of metabolizable energy [47]. This would potentially correspond to a daily additional energy supply of between 7.1% and 3.3% on average for piglet energy intake during the three consecutive treatment days.

Several studies have documented that piglet growth and death rates are closely related to piglet energy intake during the first week [19,46,48]. As there may be considerable variation in the colostrum produced and milk intake of the piglets in a litter, an additional energy supply should not be disregarded for those pigs that have the lowest energy intake. Both the test and control group had a death rate for the period below 5%, and the design of the study does not allow for a closer analysis of the importance of the extra energy supply compared to examining a reduced death rate due to the use of antibiotics.

### 4.2. Microbiology

The microbiological tests in this study were few and can at most be interpreted as indicative of potential relevant pathogens on the farm. Furthermore, no specific pathogen has conclusively been shown to be the cause of neonatal diarrhea [49]. However, several studies have shown a correlation to pathogens that individually or in combination may contribute to herd problems concerning neonatal diarrhea [13,15,49]. In this trial, the laboratory test found the following pathogens: *C. perfringens*, *E. coli*, and *Rotavirus A* (see Table 2).

*Clostridia*: Elizondo et al. (2010) found that chestnut extract inhibited various strains of *C. perfringens* with Minimum Inhibitory Concentrations (MICs) between 0.003 and 0.15 mg/mL [50]. As anaerobic cultivations at the two laboratories both indicated the presence of *Clostridia*, and as the PCR and ELISA test by Miprolab found α-toxin-producing *C. perfringens*—which after cultivation presented itself as both α- and ꞵ2-toxin-producing. *C. perfringens* may therefore have contributed to the high prevalence of neonatal diarrhea. The concentration of tannins in the intestinal lumen of piglets in the test litters was calculated to be about 0.2 mg/mL (0.27 to 0.12 mg/mL) on average when calculated on a daily basis. The product with extract from sweet chestnut could therefore be expected to reduce the need for antibiotic treatments of diarrhea in the trial litters in case the incidences truly were due to *C. perfringens*.

*E. Coli*: Microbiological analysis by both laboratories showed high numbers of non-hemolytic *E. coli* in intestinal and in fecal samples. Hemolytic *E. coli* F4/F18 species are the most commonly found types of pathogenic *E. coli* in Danish piglets [51], but these were not identified in samples from the diarrhetic piglets in the trial, even though the gilts were not vaccinated against *E. coli* carrying these virulence factors. It is possible that *E. coli* with other virulence factors than those included in the laboratory tests were involved in the diarrhetic conditions [10], or that *E. coli* caused a secondary infection which thrived and possibly exacerbated an inflammatory intestinal reaction caused by another pathogen [52]. The antibiotic applied against diarrhea targets both Gram-negative and Gram-positive bacteria and is includes *Clostridia* [53], so it is therefore not on this basis possible to establish a likely causative pathogen.

Extract from sweet chestnut has been tested as a feed additive in two trials at 1% [54] and 2% [55] (corresponding to 10 and 20 mg/mL) to curb diarrhea at weaning, where piglets were experimentally infected with entero-toxigenic *E. coli*. These trials showed that the extract reduced diarrhea and had apositive impact on both feed intake and daily growth. However, such high concentrations may not be indicative of its effect in the present trial, as the present study applied an average dosage of extract of approximately 0.2 mg/mL (0.27 to 0.12 mg/mL). On the other hand, an in vitro study which investigated extract from sweet chestnut to control *E. coli* found an MIC value of around 0.6 mg/mL depending on the strength of the growth media. The MIC was around 0.3 mg/mL at half strength of the growth media and around 1 mg/mL at one and a half the recommended strength [56]. Furthermore, bacterial strains may show substantial variation in sensitivity to tannin extracts—as illustrated by *C. perfringes* above. In this farm trial, it was not possible to conclude on the sensitivity of the identified *E. coli* to the tannin extract from sweet chestnut.

The complimentary feed tested in this farm trial also contains activated charcoal, which is known to impact *E. coli*. An in vitro trial showed that 1% (10 mg/mL) activated charcoal in growth media was able to adsorb all *E. coli*, while lower concentrations had less of an effect [57]. A review from 2019 showed a considerable variation regarding the effect of mostly long-term addition of charcoal in feed [58]. This farm trial does not allow for an evaluation of the effect of short-term application of charcoal in the feed, but it should be considered that the charcoal in the product on average contributed less than 0.2 mg/mL (0.27 to 0.12 mg/mL) to the total ingested feed amount.

*Rotavirus*: There is increasing evidence that ellagic acid, which is the main building block of the ellagitannins in the extract from sweet chestnut, has a strong inhibitory effect on several types of viruses, including *Rotavirus* [59]. Tannins and ellagic acid have very low uptake in the small intestines and may therefore be active biomolecules there. An effect of ellagic acid on *Rotavirus* should therefore not be disregarded, as the primary targets of the virus are the enterocytes of the small intestine [60,61]

In the investigated research, *Rotavirus* has often been found to be a likely cause of neonatal diarrhea [62], and the laboratory which analyzed the intestinal contents found *Rotavirus type A* in both one- and two-day-old piglets (Table 2). Especially with gilt litters, *Rotavirus* can be a problem, as primiparous sows tend to deliver less colostrum that contain a lower concentration of the specific immunoglobulin A that acts against *Rotavirus type A* after vaccination [63]. The known symptoms due to *Rotavirus* include diarrhea, vulnerability to other infections, and death for weak piglets [49,64].

Recent records from Denmark show that both *Rotavirus type A* and *C* were found in “massive” numbers in about 30% of Danish sow herds [65]. However, the laboratories used in this trial only tested for *Rotavirus type A*, and any occurrence of *Rotavirus type C* on the test farm at the time of the trial is therefore unknown. However, *Rotavirus type C* was later confirmed at a “massive” level on the farm.

### 4.3. Anti-Inflammatory and Antioxidant Physiological Effects

It is well recognized that the symptoms of diarrhea are due to inflammatory reactions in the intestinal epithelium with the release of free radicals. The resulting disruption of cells and tight junctions includes the opening of the paracellular pathway that allows water and salts to flow into the intestine [66,67,68,69]. On the other hand, chestnut extract and its two building blocks gallic acid and ellagic acid have documented anti-inflammatory and antioxidative effects [70,71,72,73] that may help in modulation, and they have a healing effect on the inflammatory reaction of the mucosa [74,75,76,77]. 

The tested product contains mono-, di-, and triglycerides of butyric acid, with an estimated average dose of 0.12 mg/mL (0.17 to 0.08 mg/mL) having been contributed to the ingested feed. Butyric acid is known to nourish the intestinal epithelium and thereby enhances intestinal barrier function [78], but butyrate also shows a certain ability to control *E. coli*, with an in vitro MIC value around 23 mg/mL and *C. perfringens* at 12 mg/mL [79]. Butyrates are well absorbed by the enterocytes, so it is therefore not clear if the short-term use of a complimentary feed with the said concentration of butyrates would contribute to the overall positive effect on diarrhea.

### 4.4. Microbiota and Immunity

The consumption of antibiotics used against other symptoms than diarrhea exhibited a statistical significant reduction of 45% (*p* = 0.045) when the test litters (16 doses) were compared to the control litters (29 doses) (Table 1).

This difference may have been caused by changes in the microbiota [80]. The commensal bacteria of the intestinal microbiota are recognized as an important component of the pig’s ability to control pathogen virulence from both Gram-negative [81] and Gram-positive bacteria [82]. In several trials and at various concentrations, extract from sweet chestnut has proven to impact the composition of the microbiota in a beneficial direction [55,83,84]. On the other hand, several studies on antibiotics have shown a short- or long-term negative influence on the composition of the microbiota [85,86,87].

Antibiotics have also been shown to have a systemic negative effect on the immune system [88] specifically its ability to control the passage of bacteria across the intestinal barrier, as was shown for several antibiotics in both short-term [88] and longer-term trials [89,90,91].

### 4.5. Antibiotic Resistance

The sampled *E. coli* exhibited a pattern of resistance to antibiotics that conformed relatively well to the general pattern found among non-pathogenic *E. coli* in Danish pigs, as documented in both independent research [92] and in the national monitoring program [34,93].

The national antibiotic resistance monitoring program, based on non-pathogenic *E. coli* sampled from pigs, covers 7 of the 15 antibiotics also tested as standard by the laboratory in Kjellerup. The *E. coli* sample from the farm shoved low resistance against three antibiotics (ciprofloxacin, colistin, and gentamicin) corresponding to a low prevalence of resistance in the national monitoring program. On the other hand, four antibiotics (ampicillin, sulfametoxazole, tetracycline, and trimethoprim) where resistance was registered in *E. coli* sampled from the farm also had a high prevalence of resistance in *E. coli* obtained from pigs on a national scale [93].

The resistance patterns for the aminoglycoside antibiotics were complex, as *E. coli* sampled at the farm showed resistance towards the aminoglycosides neomycin and streptomycin, while no resistance was registered towards gentamicin. The farm manager registered a satisfactory effect from the antimicrobial used, which contained the aminoglycoside paromomycin.

The only notable difference from the most encountered resistances in *E. coli* found in Danish pigs was the established resistance toward neomycin, as reported in 2019, wherein it only existed in *E. coli* in about 20% of Danish pig herds [92]. Research has established that resistance toward neomycin has been growing as consumption has increased [92], and it has been observed that resistance against specific antibiotics in many pathogens and indicator species in general follow the consumption of the specific antibiotic [94,95,96,97].

## 5. Conclusions

During this four-week farm trial, the consumption of antibiotics was reduced by approximately 81%, from 3.13 doses per piglet in the control group to 0.59 doses/piglet in the test group. It is concluded that the complementary feed product O-Nella-Protect—containing extract from sweet chestnut wood—in this farm trial was able to support this pig farm in meeting societal demands for the reduced consumption of antibiotics.

## Figures and Tables

**Figure 1 vetsci-12-00042-f001:**
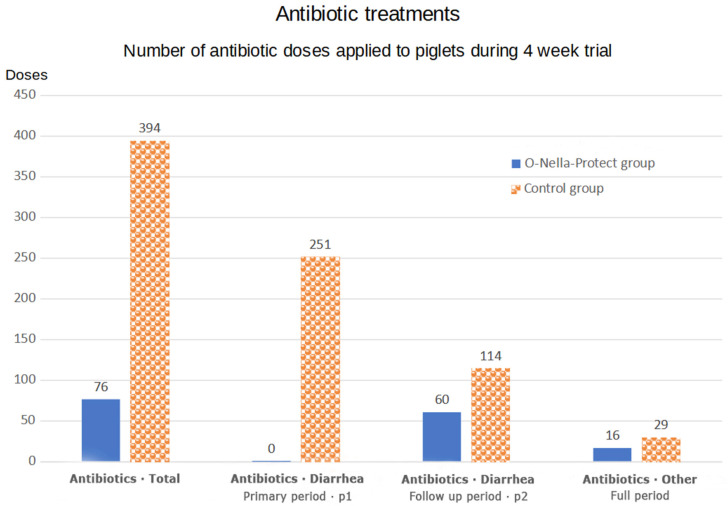
Blue columns illustrate consumption of antibiotics in Test group and orange columns illustrate consumption of antibiotics in Control group. **Left**: The total number of doses of antibiotics in test and control groups. Next; the number of doses of antibiotic doses used against diarrhea in test group and in control group during the primary treatment period (p1). Next; the number of doses used against diarrhea in test and control groups for follow up treatments (p2) **Right**: Consumption of doses of other antibiotics against all other illnesses in test group and control groups.

**Table 1 vetsci-12-00042-t001:** The number of antibiotic treatments is reported as doses of antibiotic single treatments administered to piglets in each litter.

**Test Litters (Complementary Feed Product (CFP))**											
						**Treatments**					
**Sow-no.**	**Crate no**	**Litter Size**	**Dead**	**Moved**	**Weaned**	**CFP**	**Florf.**	**Paromo.**		**Strep.**	**Total**
								**p1**	**p2**		
7376	2004/1/1	16	2	2	12	48	4	-	0	0	4
7380	2004/1/2	14	0	0	14	42	3	-	0	0	3
7387	2004/1/3	14	1	0	13	42	2	-	0	3	5
7399	2004/1/5	15	1	0	14	45	1	-	0	0	1
7377	2004/1/6	14	0	0	14	42	2	-	0	1	3
7393	2004/1/7	14	0	1	13	42	0	-	28	0	28
6811	2004/1/8	10	1	1	8	30	0	-	0	0	0
7358	2004/1/9	16	0	2	14	48	0	-	32	0	32
7398	2004/1/10	15	1	0	14	45	0	-	0	0	0
Totals		128	6	6	116	384	12	0	60	4	76
**Control Litters (Positive Control by the Use of Paromomycin)**											
						**Treatments**					
**Sow-no.**	**Crate no**	**Litter Size**	**Dead**	**Moved**	**Weaned**	**CFP**	**Florf.**	**Paromo.**		**Strep.**	**Total**
								**p1**	**p2**		
7391	2004/2/11	15	0	1	14	-	4	30	30	0	64
6805	2004/2/12	11	0	0	11	-	0	22	0	0	22
7345	2004/2/13	15	0	0	15	-	3	30	0	0	33
7374	2004/2/15	14	0	1	13	-	8	28	28	0	64
7361	2004/2/16	15	2	1	12	-	1	30	30	0	61
7362	2004/2/17	14	2	1	11	-	2	28	0	0	30
6808	2004/2/18	14	1	0	13	-	0	27	26	0	53
6813	2004/2/19	14	1	0	13	-	6	28	0	0	34
7347	2004/2/20	14	0	0	14	-	2	28	0	3	30
Totals		126	6	4	116	0	26	251	114	3	394

Guide to Treatments: CFP = complementary feed product. Florf. = florfenicol. Paromo. = paromomycin treatment, p1 first and p2 treatment periods. Strep. = Streptocillin treatment.

**Table 2 vetsci-12-00042-t002:** Microbiological analysis of pooled fecal samples at Miprolab, Germany, and from necropsy of 4 pigs at the veterinary laboratory in Kjellerup, Denmark.

Pathogen	Miprolab on Fecal Samples (Pooled)	Kjellerup on Necropsy of 4 Pigs
** *Clostridia* **	*Clostridium perfringens* α tox+ (ELISA) *Clostridium perfringens* α tox+ (PCR) *Clostridium perfringens* α&ꞵ2 tox+ (culture)	*Clostridium* sp. (Not typed) (2 samples each pooled from 2 piglets).
** *Echerichia coli* **	F4-negative, high count 6 × 10^9^	Non-hemolytic, high growth (sample of 2 piglets aged 1-day- pooled) Non-hemolytic, growth (sample of 2 piglets aged 2-days-pooled)
** *Rotavirus A* **	Negative	1-day-old piglets (pooled) Positive 2-day-old piglets (pooled) Positive

**Table 3 vetsci-12-00042-t003:** The antibiotic resistance profile of the identified *E. coli*, as reported by Kjellerup laboratory.

Antibiotic	Test Value (µg/mL)	Interpretation	Sensitive Criteria MIC ≤ (µg/mL)
Amoxicillin/clavulanic acid	16	I	8
Ampicillin	>32	R	8
Apramycin	≤4	S	8
Cefotaxime (1)	≤0.12	S	0.25
Ceftiofur (2)	≤0.5	S	2
Ciprofloxacin (3)	≤0.015	S	0.25
Colistin	≤1	S	2
Florfenicol	4	S	4
Gentamicin	≤0.5	S	4
Neomycin	>32	R	4
Spectinomycin	64	S	64
Streptomycin	128	R	8
Sulfamethoxazole	>1024	R	256
Tetracycline (4)	>32	R	4
Trimethoprim	>32	R	2

S = sensitive. I = intermediary resistance. R = resistant. (1) Cefotaxime and Ceftiofur represent cephalosporins. (2) Ciprofloxacin represents fluoroquinolones (enrofloxacin). (3) Tetracycline represents doxycycline and other tetracyclines.

## Data Availability

Detailed data from the trial are available upon request.

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
