# Peer review of "Control of Neonatal Diarrhea in Piglets with Reduced Antibiotic Use by Application of a Complementary Feed—A Randomized Controlled Farm Trial"

_vetsci, 2025, doi:10.3390/vetsci12010042_

Round 1
Reviewer 1 Report
Comments and Suggestions for Authors
The current manuscript reports the data obtained during a randomized controlled farm trial - conducted in a field setting - where a traditional antibiotic treatment was applied to newborn piglets in comparison with the utlization of a natural product based on tannins.
The work, that resembles a case-study, has sufficient originality. In a European context of antibiotic utilization reduction, and lack of other sustainable and cost-effective alternatives, studies that research new effective strategies are always welcome and interesting.
However, the current manuscript would need to undergo moderate English editing since some parts are not completely clear and could be therefore rewritten.
Moreover, there are several comments about the various sections of the paper (see below), other than a concern about the study set-up and the choice of the specific antibiotic treatment in the control group of this study, so major revisions are needed.
Section-specific comments
Introduction
The introduction lacks of a specific reference to the aim of the current study, that should be clearly stated at the end of the section.
Lines 45-46: "While farm managers WANT to avoid diarrhea among piglets, the society WANTS farmers..."
Lines 55-56: Bacterial names should be in italics.
Lines 64-71: There is only explanation and reference to tannins in this paragraph. However, in Materials and Methods it is reported that the used product also contains different forms of butyric acids. Why authors are not discussing it and mentioning the effect of this second component of the product throughout the paper, and only focus on tannins? I do think that part of the beneficial effects are also due to butyric acid, and it should be considered in all the sections of the manuscript.
Materials and methods
Lines 77-82: It is not well clear the reason why the farm is using Parofor, what is the producer, what is the active principle, what is the dosage used per pig. How is the antibiotic selected for use in this farm? Is it chosen basing on detected pathogens and their susceptibility to paromomycin, the active principle of Parofor?
Moreover, the second part of this paragraph seems more a discussion rather than an information suitable for materials and methods section.
Lines 83-89: The O-Nella-Protect producer is not reported in the paragraph, and more information about the relative abundance of each ingredient in the product are not reported. Is the butyric acid part more prevalent, or are tannins more prevalent, and at which extent?
Morevoer, the product dosage is 1 mL/piglet/day, but why is the dose compared to a general statistic of birth weight of piglets, rather than the average birth weights of piglets of the current study?
Lines 94-96: Why all controls on one side of the section, and all treatments on the other side? Wouldn't have been better to randomize also the allocation of treatments to the pens?
Lines 109-114: Doses of Parofor antibiotics are not clear. It is clear the number of "shots", but not the concentration or amount of antibiotic used for each shot.
Lines 122-128: It is not clear the selection of piglets that were not part of control or test litters, when they were samples, which samples were collected, on which samples were conducted microbiological tests, which precise tests were chosen, which pathogens were researched. Were MIC tests
Lines 142-145: two other antibiotics are mentioned: flordofen and streptocillin. Why measuring their utilization if not mentioned in all the other parts of the materials and methods section? This is not clear, also because in the results section there is a lot of data and statistical analysis presented, but they are not mentioned in materials and methods section other than explaining their statistical analysis.
Results
Lines 225-236: there is no need to keep as "supplementary" the results about the number of antibiotic treatments, the isolated bacterial strains, or their antibiotic resistance profile. I think it is worth to have them in the full article.
Discussion
Lines 246-247: the statement in these two lines is quite "strong" and should be better discussed, otherwise it could sound as an overstatement.
Lines 248-251: a few considerations are reported, but they are not well and deeply discussed in the following paragraph, with particular reference to the economic sustainability of the product utilization of this product in a commercial setting, compared to antibiotics. Considering the commercial setting of the trial, it could be worth to add some economical sustainability discussion. Moreover, there is no mention about any observed lower feed intake or digestibility in the current trial when using the plant extract. I would also suggest to discuss the possibility that free plant extracts could be degraded and/or immediately absorbed in the stomach of piglets, without reaching the intestine, where they would need to arrive in order to exert their beneficial effect.
Lines 252-261: Why is this part only considering C. perfringens, and not mentioning E. coli, like Enterotoxigenic E. coli (ETEC), that is one of the main pathogens for newborn and weaning piglets. In fact, even if F4 was not detected, there could be F18, F9, or other strains of ETEC.
Moreover, Table S2 also reports that E. coli was detected both in fecal samples and dead piglets. I do think that discussion should also include E. coli.
Lines 268-275: This paragraph should be rephrased because it is not completely clear. Moreover, authors should explain better how the astringent effect of tannins is related to the tightening of the tight junctions in the intestine.
Lines 276-282: Authors discuss that resistance towards aminoglycosides like neomycin is growing, as also reported in the antimicrobial susceptibility tests done on E. coli isolates from dead piglets. Authors should comment on the risk that E. coli strains in the farm could be resistant to aminoglycosides like Parofor. I also suggest authors to perform supplemental analysis on the bacteria isolated on fecal samples (if possible) to perform an antibiogram and understand if those isolates are resistant to the antibiotic employed in this study. In fact, one of my major concerns is that, if bacterial strains in the farm were resistant to aminoglycosides, the term of comparison of the tannin-based product (Parofor group) would not be completely appropriate and discussions would need to be rewritten.
Comments on the Quality of English LanguageAs already reported in the previous section, the current manuscript would need to undergo moderate English editing since some parts are not completely clear and could be therefore rewritten.
Author Response
Thanks for the effortthat you have done to sharpen the manuscript. They have really helped me to be more clear and systematic. Due to your many comments and questions not much of the original article has survived in untouched shape. I have therefore given up on painting the ajusted sections red in the attached revised manuscript.
We know that the tested product works for many farmers, but as the scientifi community have not been able to identify the causative pathogen it is hard to point at a single relevant mode of action.
We hope that you will be more content with this new version. And we are sorry it took so long to come back with something sensible.
Best wishes and thanks again
Klaus
The introduction lacks of a specific reference to the aim of the current study, that should be clearly stated at the end of the section.
Right - I expect that the revised version does the job better.
Comment 2: "farm managers WANT to avoid diarrhea among piglets, society WANTS farmers…"
OK OK . . got it - I think.
Comment 3: Bacterial names should be in italics.
Done that - sorry I didn´pick that up from the authors guideline.
Comment 4: There is only explanation and reference to tannins in this paragraph. However, in Materials and Methods it is reported that the used product also contains different forms of butyric acids. Why authors are not discussing it and mentioning the effect of this second component of the product throughout the paper, and only focus on tannins? I do think that part of the beneficial effects are also due to butyric acid, and it should be considered in all the sections of the manuscript.
You are completely right in raising this point. I hope the revised version will fill these gaps.
Comment 5 It is not well clear the reason why the farm is using Parofor, what is the producer, what is the active principle,
what is the dosage used per pig.
Right - its in the manuscript now.
How is the antibiotic selected for use in this farm? Is it chosen basing on detected pathogens
Well - most Danish farms really dont know what pathogens they are up against. that has several reasons - sampling takes time and often farmers have to act Now. Often they know that B. pilosicoli is this week after weaning and so on - so why not just apply that which the veterinarian suggest. When making tests you may also find things that you don´t want to find - and that may be costly. There was also some hesitation to making lab test - why should they - when the wise people at the universities couldn´t find the causative pathogen.
And Yes, the farm had abandoned the "normal" antibiotic when they experienced this diarrhetic epedemic and tested a couple before the found that this made the difference. In litters from multiparous sows they used quite another procedure because of very few diarrhetic piglets - no metaphylaxis but single animal treatment.
and their susceptibility to paromomycin, the active principle of Parofor?
Right, I hope it better presented in the revised manusript.
Comment 6: Moreover, the second part of this paragraph seems more a discussion rather than an information suitable for materials and methods section.
Right - we moved it.
Comment 7: The O-Nella-Protect producer is not reported in the paragraph, and more information about the relative abundance of each ingredient in the product are not reported. Is the butyric acid part more prevalent, or are tannins more prevalent, and at which extent?
Yes, we have trid to give more details in the revised manusript.
Comment 8: Morevoer, the product dosage is 1 mL/piglet/day, but why is the dose compared to a general statistic of birth weight of piglets, rather than the average birth weights of piglets of the current study?
The farm manager objected to any weighing of piglets. According to the farm veterinarian the gilts and their stature and health is quite comparable to average Danish pig production and he advised to use the ntaional statistics to get a sober starting point.
Comment 9: Why all controls on one side of the section, and all treatments on the other side? Wouldn't have been better to randomize also the allocation of treatments to the pens?
Yes, we wanted that but the farm manager was afraid of mistakes in their bussy daily work. She thought that it would be more safe to go rather simple.
Comment 10: Doses of Parofor antibiotics are not clear. It is clear the number of "shots", but not the concentration or amount of antibiotic used for each shot.
Right - I hope you will find the relevant information in the revised version.
Comment 11: It is not clear the selection of piglets that were not part of control or test litters, when they were samples, which samples were collected, on which samples were conducted microbiological tests, which precise tests were chosen, which pathogens were researched. Were MIC tests
The farm manager was of a very efficient mindset. we had to agree to her conditions before she would do the trial: No nothing that would take her time, we felt lucky when she agreed to have microbiological test done. MIC test on Parofor was not standard by any of the lab.
Comment 12: two other antibiotics are mentioned: flordofen and streptocillin. Why measuring their utilization if not mentioned in all the other parts of the materials and methods section? This is not clear, also because in the results section there is a lot of data and statistical analysis presented, but they are not mentioned in materials and methods section other than explaining their statistical analysis.
Yes, thanks for your systematic input also here. We hope you will find the revised version to cover subject better.
Comment 13: there is no need to keep as "supplementary" the results about the number of antibiotic treatments, the isolated bacterial strains, or their antibiotic resistance profile. I think it is worth to have them in the full article.
Thanks, we have done so, and hope the editor will agree with you.
Comment 14: the statement in these two lines is quite "strong" and should be better discussed, otherwise it could sound as an overstatement.
You are completely rigt. We have adapted it.
Comment 15: a few considerations are reported, but they are not well and deeply discussed in the following paragraph, with particular reference to the economic sustainability of the product utilization of this product in a commercial setting, compared to antibiotics. Considering the commercial setting of the trial, it could be worth to add some economical sustainability discussion.
Thanks you are completly rigtht - these considerations are not farm consideration but such that are used when designing new products. We through it out.
Danish regulation of veterinary service is rather unique in the world and therefore the economics of it is also rather unigue. No-one outside of Denmark can therefore learn much from an analysis of the economic details. However the major element of the economics is - if the piglet thrives or dies. This element seems to persuade farmers and farm mangers to use the product more and more.
Comment 16: Moreover, there is no mention about any observed lower feed intake or digestibility in the current trial when using the plant extract. I would also suggest to discuss the possibility that free plant extracts could be degraded and/or immediately absorbed in the stomach of piglets, without reaching the intestine, where they would need to arrive in order to exert their beneficial effect.
Very relevant observation. In general we see better feed intake. This is a three day treatment with tannins - and routined farm mangers prefer the product because it visibly help the piglets to thrive and have better digestion. Furthermore it would be quite difficult to distinguish between depressed feed intake due to Pigs have very high tolerance to tannins - acorn and all. Tannins and even oligo phenols have extremly low uptake. so its only when broken down to gallic acid and ellagic acid that gallic acid is absorbed to medium degree. Ellagic acid need to undergo further degradation before its various urolithin catabolites are absorbed to some degree. Most of the tannin effect on diarrhetic conditions are therfore thought to be due to MoA in the intestinal lumen and the intestinal mucosa.
Comment 17: Why is this part only considering C. perfringens, and not mentioning E. coli, like Enterotoxigenic E. coli (ETEC), that is one of the main pathogens for newborn and weaning piglets. In fact, even if F4 was not detected, there could be F18, F9, or other strains of ETEC.
Right you are. We hope you will find that the new section is more up to the task.
Comment 18: Moreover, Table S2 also reports that E. coli was detected both in fecal samples and dead piglets. I do think that discussion should also include E. coli.
You are absolutely right. Please see attached.
Comment 19: This paragraph should be rephrased because it is not completely clear. Moreover, authors should explain better how the astringent effect of tannins is related to the tightening of the tight junctions in the intestine.
We decided to leave the main part of this paragraph out. Regarding the astringent effect I have made an outline for a theoretical paper that would be able to cower documentation and theories. Thanks.
Comment 20: Authors discuss that resistance towards aminoglycosides like neomycin is growing, as also reported in the antimicrobial susceptibility tests done on E. coli isolates from dead piglets. Authors should comment on the risk that E. coli strains in the farm could be resistant to aminoglycosides like Parofor.
We have tried to discuss this and hope you will find it satisfying. As Parofor was efficient against diarrhea on a daily basis it must have worked on some pathogen/inflammation.
Comment 21: I also suggest authors to perform supplemental analysis on the bacteria isolated on fecal samples (if possible) to perform an antibiogram and understand if those isolates are resistant to the antibiotic employed in this study. In fact, one of my major concerns is that, if bacterial strains in the farm were resistant to aminoglycosides, the term of comparison of the tannin-based product (Parofor group) would not be completely appropriate and discussions would need to be rewritten.
This may be very relevant but definitely not possible at this point in time, It would take the research far beyond a farm trial and as well funded researchers at Universities in many countries struggle to identify the relevant pathogens I think its very ambitious to think that we would be able.
As we now have stipulated in the article the choise of parofor was due to its obvious effect on the diarrhea. But I get your point. We struggled to identify a farm that had real evident neonathal diarrhea - and the farm veterianarian advised us to ask this farm as he knew that gilt litters on this farm had diarrhea all ower - if no antibiotic was used.

Reviewer 2 Report
Comments and Suggestions for Authors
Title: Control of neonatal diarrhea in piglets with reduced antibiotic use by application of a complementary Feed – a Randomized Controlled Farm Trial
The manuscript describes the reduction of antibiotic consumption through application of a complimentary feed containing tannin extract to combat neonatal diarrhea in piglets. The study is both interesting and meaningful, and the manuscript is well-written.
However, it is advisable to incorporate commentary on the statistical modeling utilized in this research within the Discussion section. This practice ensures the appropriate implementation of statistical methods and facilitates readers' understanding of the underlying observations inherent in this type of experimentation.
Author Response
Comment 1 The manuscript describes the reduction of antibiotic consumption through application of a complimentary feed containing tannin extract to combat neonatal diarrhea in piglets. The study is both interesting and meaningful, and the manuscript is well-written.
Thanks for expressing your approval of the manuscript. We hope you will find the revised manuscript equally relevant.
Comment 2 However, it is advisable to incorporate commentary on the statistical modeling utilized in this research within the Discussion section. This practice ensures the appropriate implementation of statistical methods and facilitates readers' understanding of the underlying observations inherent in this type of experimentation.
We have included a new section in the discussion, highlighted in red in the attached file
Thanks again
Klaus

Reviewer 3 Report
Comments and Suggestions for Authors
Dear all,
first of all, congratulations for this nice and well-written article. I have enjoyed going through it as it is very well exposed as well as clearly written.
I might avoid using so much commericial names of both products tested- I missed the composition of Parafor.
Last section- additional material, needs to be reviewed as description of those materials contains few mistakes -necropcy instead of necropsy, a couple of dots, instead of one… minor ones and easy to be solved.
It has been a pleasure reading this article! Congrats again!
Author Response
Comment 1: First of all, congratulations for this nice and well-written article. I have enjoyed going through it as it is very well exposed as well as clearly written.
Thanks - we hope the revised version will be equally enjoyed.
Comment 2: I might avoid using so much commercial names of both products tested.
Thanks. You are absoltely rigtht and I have tried to get rid om most. Streptocillin is a special case as its a combined antibiotic. However, its not mentioned many times.
Comment 3: I missed the composition of Parafor.
Thanks, and yes - actually we have included more details on all four products that were recorded and reportet in the article in the Methods section. Its highlighted in red colour text.
Comment 4: Last section- additional material, needs to be reviewed as description of those materials contains few mistakes -necropcy instead of necropsy, a couple of dots, instead of one… minor ones and easy to be solved.
Thanks. Reviwer 1 wanted the tables moved into the text. I have tried to correct as suggested
Comment 5: It has been a pleasure reading this article! Congrats again!
And thanks again. The revised version has a much expanded discussion on the request of another reviewer. I hope it has hit the rigtht balance between exactnes and hypothesis.
Your comments were wery helpful for improving the article.
Klaus
